# Synthesis of Some Fluorescent Dyes Based on Stilbene Derivatives with Various Substituents and Their Effects on the Absorption Maxima

Yoon-Gu Lee  and Jae-Hong Choi *

Department of Textile System Engineering, Kyungpook National University, 80 DaeHak-Ro, Buk-Gu, Daegu 41566, Republic of Korea; lyg0396@naver.com
* Correspondence: jaehong@knu.ac.kr

**Abstract:** The six stilbene-based dyes containing benzoxazole substituents to improve solubility of dyes as well as the efficiency of fluorescence at blue emission were synthesized. In this work, absorption and fluorescent properties of the synthesized dyes were investigated. For the derivatization of benzoxazolyl stilbene dye, $-NO_2$ and $-NH_2$ groups were introduced in sequence onto benzoxazolyl rings. The emission maxima of the six dyes prepared were observed in the range of 435 nm~471 nm. In addition, the solubility of the dyes in dichloromethane was examined for application to the nonpolar polymer films such as PE, PP, PVC and so on. *N*-alkyl groups were determined to have a greater solubility of alkylated stilbene-based dyes than analogue containing and unsubstituted group. Furthermore, investigation of the optical effects of tortional strain according to conformation of side group was also performed. For identifying these properties, the geometry, dihedral angle, and other parameters of synthesized dyes were calculated by the density functional theory and time-dependent function using a gaussian 09 program.

**Keywords:** blue fluorescence; stilbene; benzoxazole; emission maxima; absorption maxima; dihedral angle

## 1. Introduction

Stilbene compounds consisting of heterocycles such as benzoxazole, oxadiazole and triazole are used commercially as fluorescent materials [1–3]. One of these dyes, the 4,4′-bis(2-benzoxazolyl)stilbene (BBS), is utilized as a fluorescent brightener in textiles, sensors [4], and thermochromic materials [5], which can be attributed to its high fluorescence quantum yield and a mono-exponential decay time [6]. There is a sensitive response that results in the formation of a thermally excimer in BBS dye, making it useful as a sensor or probe. Despite these optical advantages, a major shortcoming of BBS is its limited solubility in organic solvents, except for chlorinated solvents, such as tetrachloroethane ($C_2H_2Cl_4$) [6]. Moreover, similar to other fluorescent dyes, its planar structure acts as the main drawback because it causes the self-quenching and red shift in an aggregated state, such as a solid or concentrated solution state [7,8].

In this work, it was attempted to synthesize some dyes by introducing additional substituents symmetrically at the 3,3′-position of stilbene rings starting from the 4,4′-bis(2-benzoxazolyl)stilbene (BBS), which can be expected to improve solubility and luminescent properties in the aggregated state, compared to the BBS moiety. Dyes containing nitro or amino groups were prepared as precursors of the final fluorescent dyes, then *n*-butyl, 2-butyl, *n*-hexyl, anilino or pyrrolidinyl-substituted dyes were subsequently synthesized from the precursor dyes. The absorption and emission maxima of the five blue fluorescence dyes were investigated, and the solubility of each dye in the dichloromethane was also examined.

## 2. Materials and Methods

### 2.1. Chemicals and Analytic Instruments

All reactions were performed under a nitrogen atmosphere in dried glassware. The materials, reagents, and solvents were purchased from Sigma Aldrich, TCI chemical, Duksan pure chemical, and used without any additional purification.

The intermediates and fluorescent dyes were identified by HPLC (Acme 9000, Young Lin, Anyang, Republic of Korea), GC/MS spectrometry (GC-MS; 7890A-5975C GC/MSD, Agilent, Santa Clara, CA, USA), Liquid chromatography/tandem Mass spectrometry (LC-MS; Agilent, Santa Clara, CA, USA). In addition, 500 MHz $^1$H NMR (in DMSO-$d_6$ or CDCl$_3$, Advanced Digit 500, Bruker, Rheinstetten, Germany) were recorded in DMSO-$d_6$ or CDCl$_3$.

### 2.2. Computational Details

All the discrete Fourier transform (DFT) and time-dependent density functional theory (TD-DFT) calculations were performed by using Gaussian 09W software with the B3LYP, 6-31G' as the standard, which has been shown to provide accurate geometries and energies. The 6-31G' basis set was used to describe the interactions between the electrons and the core electrons of a molecule rather than 6-31G basis sets [9]. These functions improve the accuracy of the calculations for delocalized or conjugated molecules as the polarization function is added. The geometry type and core twist angle in intramolecular interactions were analyzed by geometry calculations. The twist of the main stilbene group itself and the torsional angle between the main stilbene group and the benzoxazole ring according to the presence of linear-alkyl, branched-alkyl, benzyl, alicyclic amino substituent were calculated. Hence, the Frontier molecular orbital energy calculation was performed, and the energy gaps (ΔE) between the HOMO and LUMO were also calculated from the previously described program.

### 2.3. Dye Synthesis

The synthetic scheme for the intermediates **1** and **2** is depicted in Scheme 1, and the detailed experimental conditions are described below.

**Scheme 1.** Synthetic scheme for the preparation of **intermediate 1** and **2**.

To establish a synthetic procedure for intermediates **1** and **2**, 4,4-bis(benzoxazol-2-yl)stilbene (BBS) was used as a starting material; then, two subsequent reactions, nitration and reduction reaction, were performed as shown in Scheme 1.

To prepare final fluorescent dyes, the amine groups were substituted starting from **intermediate 2** by various alkyl or aryl bromide [10], as shown in Scheme 2. Two dyes, abbreviated as **SBnBu** and **SBnHe**, were prepared using by *n*-butyl bromide and *n*-hexyl bromide to provide two carbon atom deviations. In contrast, the analogue, abbreviated as **SBnBe**, was synthesized by an S$_N$2 reaction with benzyl chloride, and the other analogue, abbreviated as **SBnPy**, was prepared by a cyclization reaction between amino nucleophile and two chlorine atoms. For the comparison with two dyes, SBnBu and SBnHe, they contained linear *N*-alkyl substituents; iso-butyl group was introduced into two amine groups of **intermediate 2**, providing dye **SB2Bu**.

**Scheme 2.** Synthesis of the six fluorescent dyes starting from the **intermediate 2**.

### 2.3.1. Synthesis of Dye Intermediate 1: Nitration [11]

4,4′-bis(2-benzoxazolyl)stilbene (BBS) (20 mmol, 8.28 g) was added to sulfuric acid (80 mL) followed by stirring the mixture at 45 °C for 1 h; then, the mixture was cooled in an ice bath. The mixed acid (1:1 weight ratio of c-HNO$_3$ to c-H$_2$SO$_4$, 40 mmol; 7.79 g) was slowly added at 0–5 °C temperature followed by a stirring until the solution became transparent. When the reaction was complete, reaction solution was poured over ice to obtain crude product as a precipitated solid. The pH of crude product was then adjusted to pH = 7~8 by a slow addition of sodium hydroxide pellets in water, and the precipitated product was isolated by vacuum filtration and washed with sufficient water (500 mL). The solid was recrystallized in acetone (500 mL) to afford 1,2-bis(4-(benzoxazolyl)-3-nitro)stilbene (9.70 g, 86%) as yellow flake-like crystals. $^1$H NMR (500 MHz, DMSO-d$_6$), δ: 6.56 (m, 6H, ArH), 6.80 (t, 4H, CH·ArH), 7.01 (d, 2H, ArH), 8.62 (s, 1H, ArH), 8.69 (s, 1H, ArH), 8.86 (s, 1H, ArH), 8.93 (s, 1H, ArH); LCMS (*m/z*): [M + H]$^+$ calculated for C$_{28}$H$_{16}$N$_4$O$_6$: 504.11; determined 504.40.

### 2.3.2. Synthesis of Stilbene-Based Dye Intermediate 2: Reduction [11]

**Intermediate 1** (19.2 mmol, 9.70 g) and hydrochloric acid (100 mL) were added to a 3-neck round-bottom flask. The mixture was stirred at 50 °C for 1 h until it was well-dispersed. Then, tin (II) chloride hydrate (96 mmol, 18.20 g) was slowly added, changing the solution to an orange color. The reaction mixture was continuously stirred at 60 °C for 45 min. After cooling, the pH was adjusted to pH = 8 with the addition of aq. NaOH solution. The precipitated solid upon neutralization was isolated by vacuum filtration and recrystallized in DMF (200 mL) to obtain 1,2-bis(4-(benzoxazolyl)-3-amino)stilbene. The product was further purified by column chromatography (adsorbent: silica gel, eluent: ethyl acetate/hexane (1:1)). Yield: 50.4%; $^1$H NMR (500 MHz, DMSO-d$_6$), δ: 4.43 (s, 4H, NH$_2$), 5.28 (s, 2H, ArH), 5.57 (s, 2H, ArH), 6.05 (s, 2H, CH), 6.18 (d, 2H, ArH), 6.24 (d, 2H, ArH), 6.30 (s, 2H, ArH), 6.40 (dd, 2H, ArH), 7.53 (s, 2H, ArH); LCMS (*m/z*): [M + H]$^+$ calculated for C$_{28}$H$_{20}$N$_4$O$_2$: 444.16; determined 444.51.

### 2.3.3. Synthesis of 5,5′-(Thane-1,2-diyl)bis(2-(benzo[d]oxazol-2-yl)-N-butylaniline) SbnBu

Potassium carbonate (68 mmol, 0.94 g) was added to DMF (30 mL) followed by intermediate 2 (23 mmol, 1 g), and the mixture was stirred at 60 °C for 1 h. Then, 1-bromobutane (69 mmol, 0.95 g) was added to the reaction mixture, and stirring continued for 8 h. Once the reaction was complete, the solution was poured into water, and the resulting crystals were isolated by vacuum filtration. The crystals were further purified by recrystallization using DMF. The product was further purified by column chromatography (adsorbent: silica gel, eluent: methanol/dichloromethane (5:95)). The final product in 48.1% yield: $^1$H NMR (500 MHz, DMSO-d$_6$), δ: 0.88 (m, 4H, CH$_3$), 1.01 (s, 2H, CH$_3$), 1.42 (m, 6H, CH$_2$), 1.80 (dd, 4H, CH$_2$), 3.00 (dd, 4H, CH$_2$), 5.35 (s, 2H, ArH), 5.59 (d, 2H, ArH), 6.13 (s, 2H, CH), 6.21 (s, 2H, ArH), 6.22 (s, 2H, NH), 6.35 (d, 2H, ArH), 6.47 (d, 2H, ArH), 7.74 (d, 2H, ArH); LCMS (*m/z*): [M + H]$^+$ calculated for C$_{36}$H$_{36}$N$_4$O$_2$: 556.28; determined 556.55.

### 2.3.4. Synthesis of 5,5′-(Ethene-1,2-diyl)bis(2-(benzo[d]oxazole-2-yl)-N-hexylaniline) SBnHe

SBnHe was synthesized using **intermediate 2** (23 mmol, 1 g) following the same synthetic method used for SBnBu, except with 1-bromohexane (69 mmol, 1.14 g) instead of 1-bromobutane. The final product was obtained in a yield of 47.0%. $^1$H NMR (500 MHz, DMSO-d$_6$), δ: 0.75 (m, 4H, CH$_3$), 0.89 (d, 2H, CH$_3$), 1.23 (t, 4H, CH$_2$), 1.33 (m, 8H, CH$_2$), 1.81 (dd, 4H, CH$_2$), 2.97 (dd, 4H, CH$_2$), 5.33 (s, 2H, ArH), 5.62 (s, 2H, ArH), 6.13 (d, 2H, CH), 6.22 (dd, 4H, NH·ArH), 6.34 (dd, 4H, ArH), 6.42 (d, 2H, ArH), 7.73 (d, 2H, ArH); LCMS (*m/z*): [M + H]$^+$ calculated for C$_{40}$H$_{44}$N$_4$O$_2$: 612.82; determined 612.88.

### 2.3.5. Synthesis of 5,5′-(Ethene-1,2-diyl)bis(2-(benzo[d]oxazole-2-yl)-N-benzylaniline) SBnBe

**Intermediate 2** (23 mmol, 1 g) was dissolved in DMF (30 mL), and then potassium carbonate (68 mmol, 0.94 g) and benzyl chloride (69 mmol, 0.87 g) were added. The mixture

was heated to 60 °C and stirred for 8 h. Once the reaction was completed, the mixture was poured into water to form crystals upon standing at room temperature, then solids were isolated under vacuum filtration. The crystals were recrystallized using DMF. The product was further purified by column chromatography (adsorbent: silica gel, eluent: methanol/dichloromethane (5:95)), affording the final product in 56.1% yield. $^1$H NMR (500 MHz, DMSO-d$_6$), δ: 4.32 (d, 2H, CH$_2$), 4.91 (d, 2H, CH$_2$), 6.07 (d, 2H, ArH), 6.31 (d, 2H, NH), 6.67 (m, 6H, ArH), 6.85 (dd, 8H, ArH), 7.13 (t, 2H, CH), 7.31 (t, 2H, ArH) 8.21 (d, 2H, ArH); LCMS (*m/z*): [M + H]$^+$ calculated for C$_{40}$H$_{28}$N$_4$O$_2$: 696.69; determined 696.79.

### 2.3.6. Synthesis of 5,5′-(Ethene-1,2-diyl)bis(2-(benzo[d]oxazole-2-yl)-N-cyclopentylaniline) SBnPy

**Intermediate 2** (23 mmol, 1 g) and potassium carbonate (68 mmol, 0.94 g) were added to DMF (30 mL) followed by stirring the mixture at room temperature for 2 h until it was sufficiently dissolved. 1,4-dibromobutane (69 mmol, 1.49 g) was added, and the reaction was continuously = stirred at 100 °C for 1 hr. After the mixture was allowed to cool, it was poured into water, resulting in the formation of crystals upon standing at room temperature. The formed solids were isolated under vacuum filtration, and the crystals were recrystallized using DMF. The product was further purified by column chromatography (adsorbent: silica gel, eluent: Acetone/hexane(1:1)). The final product in 77.2% yield. $^1$H NMR (500 MHz, DMSO-d$_6$), δ: 1.82 (d, 4H, CH2), 1.94 (d, 4H, CH2), 3.45 (d,2H, CH2), 3.89 (d, 2H, CH$_2$), 4.55 (t, 2H, CH$_2$), 5.39 (d, 2H, CH$_2$), 5.82 (d, 2H, CH), 6.12 (dd, 6H, ArH), 6.31 (dd, 4H, ArH), 6.39 (t, 2H, ArH), 8.18 (t, 2H, ArH) LCMS (*m/z*): [M + H]$^+$ calculated for C$_{36}$H$_{32}$N$_4$O$_2$: 552.25; determined 552.61.

### 2.3.7. Synthesis of 5,5′-(Ethene-1,2-diyl)bis(2-(benzo[d]oxazol-2-yl)-N-(sec-butyl)aniline) SB2Bu

SB2Bu was synthesized using **intermediate 2** (23 mmol, 1 g) based on the SBnBu synthetic method using 2-bromobutane (69 mmol, 0.95 g) instead of 1-bromobutane, affording the final product in 55.9% yield. $^1$H NMR (500 MHz, DMSO-d$_6$), δ: 0.47 (t, 6H, CH$_3$), 1.02 (t, 6H, CH$_3$), 1.21 (s, 2H, CH$_2$), 1.73 (d, 2H, CH$_2$), 3.44 (d, 2H, CH), 5.29 (s, 2H, NH), 6.19 (s, 2H, ArH), 6.38 (d, 2H, ArH), 6.68 (s, 2H, CH), 7.06 (d, 6H, ArH), 7.28 (d, 2H, ArH), 8.14 (d, 2H, ArH); LCMS (*m/z*): [M + H]$^+$ calculated for C$_{36}$H$_{36}$N$_4$O$_2$: 556.28; determined 556.66;

### 2.3.8. Synthesis of (E)-N,N′-(Ethene-1,2-diylbis(6-(benzo[d]oxazol-2-yl)-3,1-phenylene)) diacetamide SBnAc

SBnAc was synthesized under solvent-free condition using **intermediate 2** (23 mmol, 1 g). Acetic anhydride (Excess, 10 mL) and **intermediate 2** were stirred at room temperature for 5 h. When the reaction was completed, the remaining liquids were removed by a rotary evaporator. The product was then purified by column chromatography (adsorbent: silica gel, eluent: Acetone/n-hexane (1:1)). $^1$H NMR (500 MHz, DMSO-d$_6$), δ: 1.27 (s, 1H, CH3), 1.35 (s, 1H, CH3), 1.72 (d, 2H, CH3), 1.88 (s, 1H, CH3), 2.06 (s, 1H, CH3), 6.32 (d, 2H, ArH), 6.49 (d, 2H, ArH), 6.78 (d, 7H, ArH), 6.97 (s, 2H, ArH), 7.89 (d, 2H, ArH), 8.07 (s, 1H, NH), 9.04 (s, 1H, ArH), 9.23 (s, 1H, ArH); LCMS (*m/z*): [M + H]$^+$ calculated for C$_{32}$H$_{24}$N$_4$O$_4$: 528.18; determined 528.61.

### *2.4. Absorption and Fluorescent Properties*

UV–Visible spectrophotometry was carried out by dissolving samples in $5.0 \times 10^{-5}$ mol/L of toluene (Thermo Scientific, Evolution 300), and fluorescent spectra were recorded on a fluorescence spectrophotometer (Scinco, FluoroMate FS-2) with the samples dissolved in $5.0 \times 10^{-5}$ mol/L of toluene. PMT voltage was measured by the fluorescent intensity at 500, and quantum yields were calculated using anthracene (Φ = 0.27) as a reference for blue fluorescence. All fluorescent properties were measured at the excitation wavelength corresponding to the maximum absorption wavelength. The quantum yields were determined with anthracene as a standard dye and measured by dividing into two parts, a dilute

solution ($1.0 \times 10^{-5}$ mol/L) and a concentrated solution ($1.0 \times 10^{-3}$ mol/L), in toluene in order to examine the influence of dye concentration on the luminescence.

## 3. Results and Discussion

### 3.1. Design Concept of Dye Molecule and Synthesis of Dyes

Although 4,4′-bis(2-benzoxazolyl)stilbene (BBS) molecule has good thermal stability and exhibits remarkable optical properties, the compound possesses poor solubility in nonpolar organic solvents, owing to its high degree of symmetry and extensive conjugated system, leading to ready molecular stacking. For the same reason, quenching in the dye's solid state occurs readily. Therefore, all the novel dyes of this study were designed to introduce *N*-substituents that confer high solubility in organic solvents as well as to reduce the red-shift behavior that derived from the excimer formation in the solid state of BBS. In addition, the effects of a strong electron-withdrawing group (-NO$_2$) and a strong electron-donating group (-NH$_2$) on the spectral properties of dye molecules were compared. One of main purposes of this study was to improve quantum efficiency of fluorescence by introducing steric hindrance through *N*-substitution with bulky functions, such as an iso-butyl group. In similar respect, to compare the effects of carbon length of *N*-alkyl groups, *n*-butyl and *n*-hexyl groups were introduced. In addition, the effect of cyclic functions, such as phenyl or pyrrolidine groups, on the spectral properties was investigated from the perspective of aromatic/non-aromatic substituents and torsional strain. Also studied was the impact of an *N*-acetyl group, introduced from the precursor containing amino groups to afford **SBnAc** dye with different hydrogen-bonding characteristics.

In the preparation of dye **intermediate 1**, a nitration reaction was carried out under low temperature of 0–5 °C, resulting in reaction yield of 86.4%. As is the tendency with nitration reactions, higher temperatures caused the formation of larger amounts of side products, thereby dramatically decreasing yield.

For *N*-alkylation reaction, the use of polar aprotic solvent, such as DMF, could enhance the reactivity of typical S$_N$2 reaction [12]. In S$_N$2 reactions between primary amines and alkyl halides, the formation of mono-alkylated products is more favorable than that of di-alkylated side-products; however, di-alkylation is inevitable. Thus, to increase selectivity toward generation of mono-alkylated compounds, the control of reaction rate was necessary in the reaction stage, resulting in reaction yield range of 48–56%. Two alkylated analogues (**SBnBu** and **SB2Bu**) were selectively prepared from corresponding alkyl halide (1-bromobutane, 2-bromobutane), whereas the cyclic group substituted analogues (**SBnBe** and **SBnPy**) were prepared from benzyl bromide and 1,4-dibromobutane.

The **SBnBu**, **SBnHe**, and **SB2Bu** dyes contain hydrophobic *N*-alkyl groups located symmetrically on the central phenyl rings of the stilbene motif, thus providing solubility. In addition, it was intended that these substituents introduce steric strain to the ring of the stilbene and benzoxazole group with the aim of reducing self-quenching as well as improving solubility in organic solvents. Of special interest, in the case of **SB2Bu** which contains bulky groups compared to that of **SBnBu**, were the effects of different bulkiness with the same carbon number.

### 3.2. Geometry Optimization of Synthesized Dyes

The geometry-optimized structures of the synthesized dyes, which were obtained by DFT calculations using Gaussian09, are shown in Figure 1. The planarity of the dye molecule can be visualized in these structures where it could be assumed that the properties of substituents mainly affected the interactions between adjacent molecules and their fluorescence properties. As shown in **(a)** of Figure 1, the two phenyl rings around the ethylene linkage are twisted by 18° out of plane, whereas the analogue dye **SBnHe** was free of angular strain (0°) by an ethylene linkage, as seen in **(b)**, indicating that longer length of alkyl chain reduced angular strain within the core moieties which could be attributable to the larger interactions between a hexyl chain and an ethylene group [13]. Similarly, this effect was observed in **SB2Bu** dye containing 2-butylamino substituents. They provided

smaller surface areas due to the more spherical than straight-chain alkanes; thus, the greater twist angles were seen, such as 32°, shown in **(e)**, for **SB2Bu**. In the case of **SB2Bu**, the less linear nature of the 2-butylamino substituents leads to greater hindrance and thus furnishes increased angle of twist of 32° as shown in **(e)**.

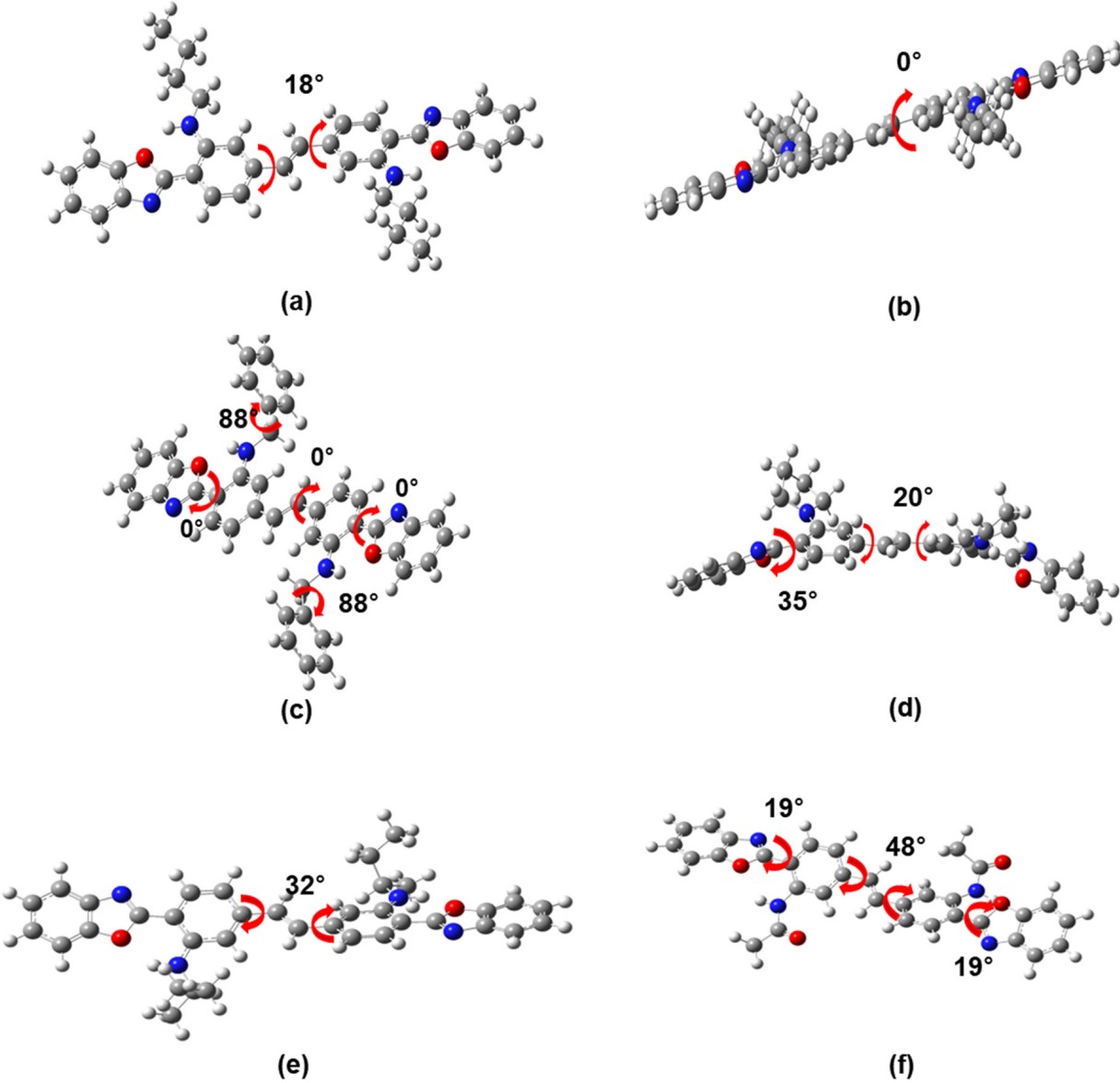

**Figure 1.** The torsion angle of minimum energy geometry of (**a**) SBnBu, (**b**) SBnHe, (**c**) SBnBe, (**d**) SBnPy, (**e**) SB2Bu, (**f**) SBnAc.

As shown in **(c)** and **(d)**, much higher planarity of the anilino groups and benzoxazole rings was observed for analogue **SBnBe**, which can be explained by the increased π-conjugation throughout the phenyl and benzoxazole systems. Therefore, the twist angles between a benzoxazole and a phenyl ring were calculated to be 0° for **SBnBe**, whereas the corresponding angles were 35° for **SBnPy**. It is also of note that **SBnPy** dye was influenced primarily by its pyrrolidinyl groups to be in twisted form of an 'X' shape, as shown in **(d)**.

The presence of acetamide groups in **SBnAc** led to significant steric hindrance as evidenced by a twist angle of 48° between phenyl ring and ethylene linkage of the stilbene moiety, as depicted in Figure 1f, in addition to a dihedral angle between the benzoxazole system and phenyl substituent of 19°.

### 3.3. Absorption and Fluorescent Properties of Synthesized Dyes

The six synthesized dyes are consistently hypsochromically shifted compared to the starting molecule (BBS) which contains no substituent. The introduction of two electron-donating groups, such as *N*-alkyl, anilino or pyrrolidine, readily destabilizes the LUMO by increasing the electron density in the LUMO, leading to an increased energy gap (ΔE) between HOMO and LUMO. For instance, as in the Figure 2, the presence of *n*-hexyl substituents gives rise to a lower energy LUMO level (−2.2 eV for **BBS**, −1.97 eV for **SBnHe**); therefore, the calculated ΔE increased from 3.45 eV to 3.71 eV, leading to the shorter wavelength absorption maximum of **SBnHe**.

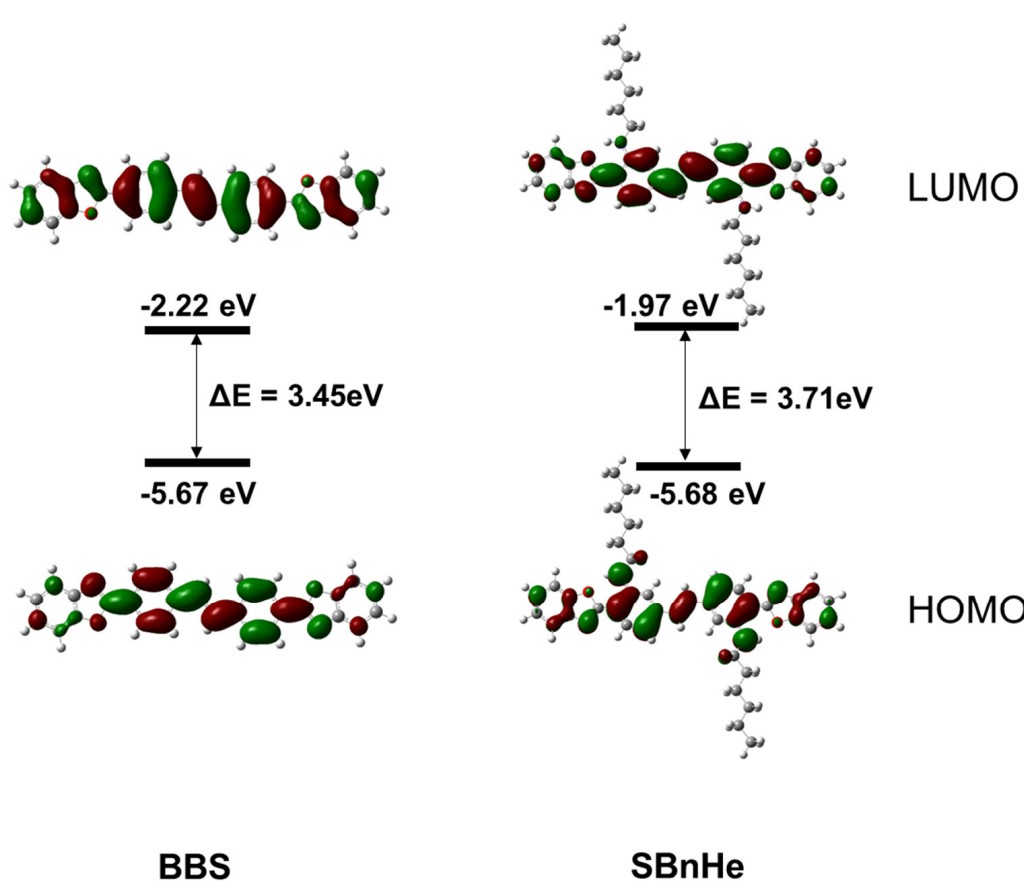

**Figure 2.** Frontier orbital models of the 4,4′-bis(2-benzoxazolyl)stilbene and **SBnHe**.

Table 1 shows that the introduction of an electron-withdrawing group (-NO$_2$) onto phenyl ring affords a bathochromic shift in absorption maximum compared to BBS, whereas a hypsochromic shift is caused by the presence of an electron-donating group (-NH$_2$). Therefore, the **intermediate 1** absorbs maximally at 410 nm, which constitutes a red shift of 35 nm from the absorption maximum of BBS, whereas **intermediate 2** exhibits an absorption maximum of 353 nm and a blue shift of 22 nm. As it was described previously, the absorption maxima were greatly affected by the electronic properties of additional substituents that could be introduced to the phenyl rings. Absorption maximum is thus greatly affected by the electronic properties of substituents attached to the phenyl rings of BBS.

The three analogous dyes **SBnBu**, **SBnHe**, and **SB2Bu** which contain *N*-alkyl groups exhibited very similar absorption maxima in the range of 350 nm and 352 nm; thus, it can be concluded that the carbon length or atom arrangement of *N*-alkyl substituent at the *ortho*-position of the carbon substituted by a benzoxazole ring did not affect the energy difference between HOMO-LUMO states. In contrast, the dyes containing cyclic groups, such as *N*-phenyl or pyrrolidinyl groups, showed different characteristics. **SBnBe**

substituted by *N*-phenyl group exhibits an absorption maximum of 360 nm, a 15 nm hypsochromic shift with respect to BBS, whereas **SBnPy** substituted by pyrrolidinyl groups absorbs maximally at 325 nm and is thus hypsochromically shifted by 50 nm compared to BBS. In the case of **SBnBe**, the increased π-conjugation between two phenyl rings via an –NH- linkage could minimize the hypsochromic shift leading to the compensation of the hypsochromic effect exerted by electron-donating groups [14]. For **SBnPy**, the combination of bulk and a lack of aromaticity of its pyrrolindo substituents therefore afforded a dramatic blue shift which is presumably a consequence of the highly twisted bond angle (35°) between benzoxazole and phenyl systems causing decreased π-conjugation leading to an additional energy difference between HOMO and LUMO. The absorption maxima of the two intermediates and six fluorescent dyes are summarized in Table 1. In the case of dye **SBnAc**, the absorption maximum was largely blue-shifted (33 nm) compared to that of corresponding **intermediate 2**. It can be assumed that the introduction of acetyl groups to the amino substituents should provide more electron withdrawing effect than the intermediate 2. However, the large twist angles (48°), as shown in Figure 1, may counteract the electronic effect to result in a bathochromic shift instead.

**Table 1.** Absorption and fluorescent properties of the synthesized dyes.

| Dyes | $\lambda_{max}^{ab}$ (nm) | $\lambda_{max}^{em}$ (nm) | Stokes' Shift (nm) | $\Phi_f^a$ | $\Phi_f^b$ |
|---|---|---|---|---|---|
| BBS | 375 | 431 | 56 | 0.86 | 0.17 |
| Intermediate 1 | 410 | 513 | 103 | 0.26 | 0.26 |
| Intermediate 2 | 353 | 472 | 119 | 0.28 | 0.11 |
| SBnBu | 350 | 458 | 108 | 0.68 | 0.66 |
| SBnHe | 352 | 460 | 108 | 0.36 | 0.30 |
| SBnBe | 360 | 465 | 105 | 0.66 | 0.61 |
| SBnPy | 325 | 471 | 146 | 0.18 | 0.15 |
| SB2Bu | 351 | 447 | 94 | 0.72 | 0.71 |
| SBnAc | 320 | 435 | 115 | 0.93 | 0.90 |

$\lambda_{max}^{ab}$, $\lambda_{max}^{em}$: determined in toluene $(1.0 \times 10^{-5}$ mol/L). Quantum yield reference: Anthracene $(\Phi = 0.27)$. $\Phi_f^a$ =quantum yield in dilute solution $(1.0 \times 10^{-5}$ mol/L). $\Phi_f^b$ = quantum yield in concentrated solution $(1.0 \times 10^{-3}$ mol /L).

In the six synthesized dyes, the longest emitter is **SBnPy** at 471 nm, whereas analogue dye **SB2Bu** emitted at shortest wavelength of 447 nm, as shown in Table 1. Generally, the alkylamino-substituted dyes provide large Stokes shifts, because the presence of electron-donating alkylamino groups stabilizes the excited state of the molecule. The same properties were reported in case of pyrrolidinyl or benzylamino groups substituted dyes [15,16]. All of the *N*-substituted dyes exhibit larger Stokes' shifts in the range of 94 nm~146 nm compared to that of BBS (56 nm); thus, all the fluorescent dyes prepared emitted either a neutral blue or a greenish-blue fluorescence, as shown in Figure 3. The smallest Stokes shift among the set of dyes was that of **SB2Bu**, which contains 2-butylamino substituents providing a comparatively higher energy gap between $S_1$ and $S_0$ states after a vibrational relaxation which may be caused by its highly twisted bond angle between phenyl ring and ethylene group, as shown in Scheme 2. In contrast, **SBnPy** containing pyrrolidinyl groups gave rise to the largest Stokes's shift of 146 nm that was unusually larger than that of analogue **SB2Bu**. The resulting shift is presumably a consequence of the greater twist bond angle (35°) between benzoxazole and phenyl systems, leading to enhanced vibration relaxation from $S_1$ state. The other three dyes **SBnBu**, **SBnHe**, and **SBnBe** showed very similar values in Stokes' shift, 108 nm, 108 nm and 105 nm, respectively, caused by their greater planarity, as seen in Scheme 2. It is of note that the unsubstituted dye **BBS** exhibits the smallest Stokes' shift of 56 nm in this series, which may be attributable to its planar structure.

Five of the six synthesized dyes exhibit lower quantum yields than that of the unsubstituted dye (BBS) in dilute solution, as shown in Table 1 and Figure 3. **SBuAc**, however, does not. This observation may be associated with the alkyl or cyclic chains exerting a greater

effect on non-radiative channels than on radiative emission. It also may be that the longer alkylamino chains introduced to the dye **SBnHe** enhanced the probability of non-radiative processes, such as a vibrational relaxation and an intersystem crossing, which may be due to the greater molecular motions and interactions with the surrounding environments. As a result of these processes, the efficiency of fluorescent emission decreased, leading to a reduction in the quantum yields of the dyes prepared. The lowest fluorescent efficiency (quantum yield 0.18 or 0.15) was detected with the dye **SBnPy** containing pyrrolidinyl groups, which can be associated with the increase in the non-radiative channel caused by the flexible alicyclic rings interacting with the rest of the dye molecule.

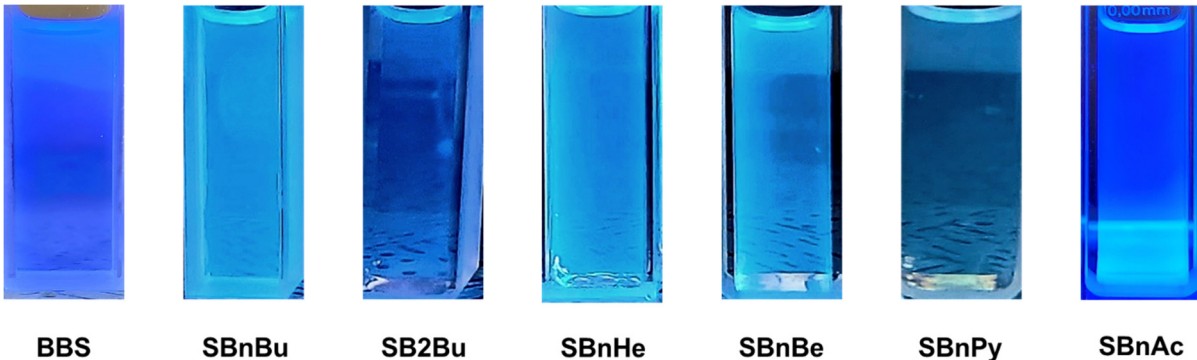

**Figure 3.** Photographs of synthesized dyes in dilute solution (toluene) under a UV light of 365 nm.

Furthermore, it was observed that the dyes substituted by a shorter butyl group (**SBnBu, SB2Bu**) or aromatic substituent (**SBnBe**) exhibited much higher quantum yields. This effect has been reported in the literature and highlights the importance of considering the impact of the length and isomerism of the alkyl substituents on the fluorescence properties of the dyes [17–23]. In terms of the torsion angles, as illustrated in Figure 1, **SBnBe** and **SB2Bu** are more distorted with respect to the stilbene moiety and exhibit higher fluorescent efficiency than the planar analogue **SBnHe**.

The most striking results were obtained in the dye **SBnAc** containing *N*-acetyl groups. It possesses greater luminescence efficiency than that of corresponding **BBS**, both in dilute and concentrated solution. In dilute solution, **SBnAc** shows a quantum yield of 0.93 compared to 0.86 for BBS. In concentrated solution, the difference in efficiency becomes much larger (0.90 for **SBnAc**, 0.17 for BBS). The enhanced luminous efficiency of **SBnAc** could be attributable to the highly twisted structure throughout the molecule, as previously described in the discussion part of geometry optimization.

As was mentioned in the introduction, the large decrease in quantum yield of BBS in the concentrated solution is likely caused by self-quenching, accompanied by red-shifted emission, as shown in Figure 4 and Table 1 [24], where the quantum yield of BBS was reduced from 0.86 in dilute solution to 0.17 in concentrated solution. It can be assumed that BBS readily undergoes ACQ process leading to significant self-quenching and red shifting of fluorescence owing to energy being transferred to the other dye molecules rather than being emitted as a luminescence. Extent and type of aggregation can significantly change the luminescence behavior of molecules, as shown in Figure 4. BBS molecules exhibit a red-shifted emission spectrum in the aggregated state with head-to-tail stacking with slip stack of C-H⋯π; as has been quantum-simulated in the literature, it seems to be a form of J-aggregation type [5]. A similar red shift was reported in the PVDF film with the higher wt % of BBS [25]. It was also reported that the aggregated dye molecules generate a new excitonic band by energy splitting, leading to a red shift in both absorption and emission spectra with respect to the monomeric analogue [26,27]. In contrast, aggregation-induced self-quenching occurred marginally for the synthesized dyes depending on the dye concentrations, thus very small reductions in quantum yields for the six dyes were observed (0.01–0.06) in the concentrated solution in comparison with that in the dilute solution. Therefore, it can be

emphasized that the inherent drawback of BBS molecule can be overcome sufficiently by the introduction of further substituents in the phenyl rings, particularly for application areas requiring high concentration of BBS-based fluorescent dyes.

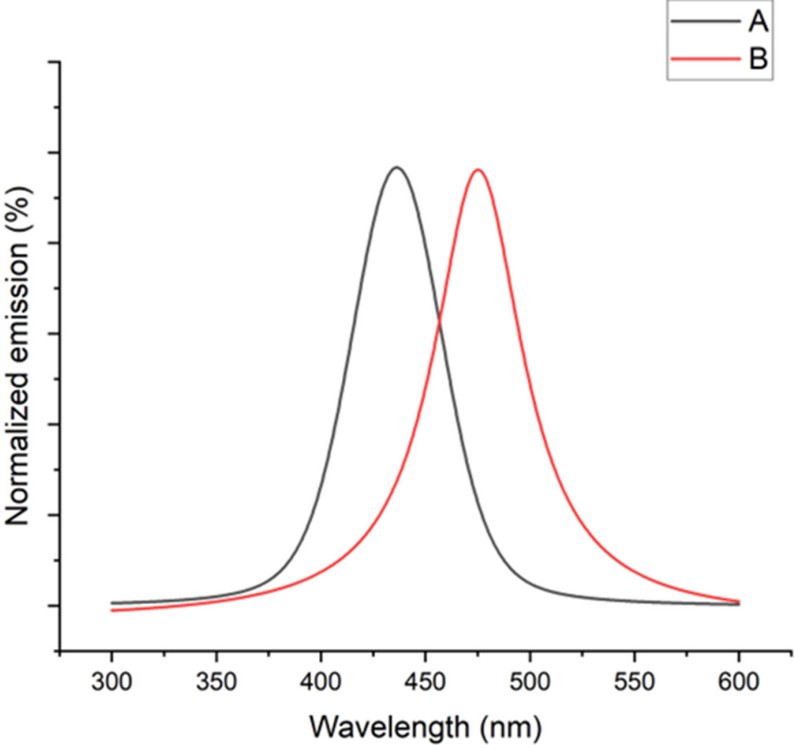

**Figure 4.** Red-shifted emission wavelength of the BBS in (A) dilute solution ($1.0 \times 10^{-5}$ mol/L). (B) In concentrated solution ($1.0 \times 10^{-3}$ mol/L).

The normalized emission spectra of the six fluorescent dyes are illustrated in Figure 5.

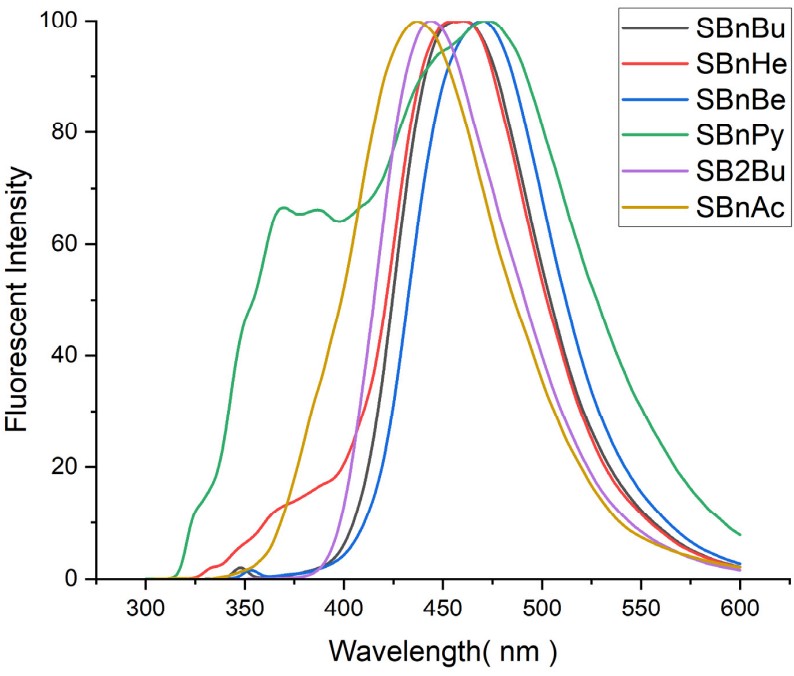

**Figure 5.** Normalized emission spectra of the synthesized dyes.

Photographs of the synthesized dyes in toluene solution are illustrated in Figure 3.

*3.4. Solubility Tests*

There are various factors, such as the polarity of molecule, π–π interactions in dyes themselves, intermolecular hydrogen bonding with the solvent, which directly affect the solubility of the dye. In the case of π–π interactions, it can be increased by the extension of the p orbital conjugation as the distance between the molecules shrinks BBS, which has a planar highly conjugated structure; therefore, it shows a strong tendency to aggregate, leading to poor solubility in most organic solvents. As shown in Table 1, BBS has a solubility of 0.3 mmol/L in $CH_2Cl_2$, which is not satisfactory for many applications, for example, light conversion films, OLEDs and other electronic materials.

The prepared dyes which have alkyl and aniline substituents exerted increased solubility in dichloromethane, as shown in Table 2. The two isomers **SBnBu** and **SB2Bu** possess similar solubilities (11.1 and 9.9 mmol/L, respectively), indicating the little impact of the shape of butyl substituents on the solubility of these dyes. Extending the alkyl chain leads to increased solubility of 14.1 mmol/L for **SbnHe**, owing to enhanced hydrophobicity from *n*-hexyl groups. The hydrophobicity of the phenyl substituents of **SBnBe** contributes to its solubility, reaching 10.0 mmol/L. In SBnAc, the presence of the acetamide group lowered solubility in non-polar solvents to 4.61 mmol/L. Since the C=O group and the N-H group are polar, it was expected that their presence would result in more significant reduction in solubility compared to other synthesized dyes. The relevant results of dye solubility are summarized in Table 2.

**Table 2.** Solubility of the synthesized dyes in methylene chloride.

| Dyes | Solubility (mmol/L) |
|:---:|:---:|
| **BBS** | 0.3 |
| **SBnBu** | 11.1 |
| **SB2Bu** | 9.9 |
| **SBnHe** | 14.1 |
| **SBnBe** | 10.0 |
| **SBnAc** | 4.6 |

**4. Conclusions**

As a series of Bis(benzoxazol-2-yl) stilbene-based fluorescent dyes, two intermediates and six corresponding dyes were prepared, then their absorption and fluorescence maxima were analyzed, the solubility providing the following conclusions.

In terms of the absorption maxima, the introduction of *N*-alkyl, *N*-acetyl, or *N*-cyclic groups to the phenyl rings consistently provided a hypsochromic shift to the BBS, whereas the presence of the nitro groups exerted a bathochromic shift.

Five of the six synthesized dyes exhibited lower quantum yields than the analogue dye BBS in dilute solution; the exception was SBuAc. The fluorescence quantum yield of BBS was dramatically decreased in concentrated solution, which was in obvious contrast to the other six dyes prepared. It was concluded that the introduction of substituents onto the phenyl rings of BBS greatly contributed to higher quantum yields in concentrated solution. In particular, the presence of *N*-acetyl groups afforded higher efficiency both in dilute and concentrated solution, which can be mainly correlated to the resultant highly twisted molecular structure. As was expected, the solubility of some dyes prepared was greatly increased in comparison with that of BBS, which is attributable to the introduction of more hydrophobic substituents.

**Author Contributions:** All authors contributed for the preparation of this manuscript. Conceptualization, Y.-G.L. and J.-H.C.; validation, Y.-G.L. and J.-H.C.; formal analysis, Y.-G.L.; data curation, Y.-G.L.; writing—original draft preparation, Y.-G.L.; writing—review and editing, J.-H.C.; visualiza-

tion, Y.-G.L.; supervision, J.-H.C.; project administration, J.-H.C.; funding acquisition, J.-H.C.; All authors have read and agreed to the published version of the manuscript.

**Funding:** This research was funded by the Ministry of Trade, Industry and Energy, grant number 10052798 "Development of organic photo-functional materials and light conversion films that enhance blue and red sun light through light conversion for improved agricultural quality and productivity".

**Institutional Review Board Statement:** Not applicable for this manuscript.

**Informed Consent Statement:** Not applicable.

**Data Availability Statement:** Further data are available from the corresponding author on request.

**Acknowledgments:** This work was supported by the Technology Innovation Program funded by the Ministry of Trade, Industry and Energy (Republic of Korea) (10052798).

**Conflicts of Interest:** The authors declare no conflict of interest.

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
