# Peer review of "Synthesis of Some Fluorescent Dyes Based on Stilbene Derivatives with Various Substituents and Their Effects on the Absorption Maxima"

_applsci, doi:10.3390/app13095543_

Round 1

Reviewer 1 Report

Lee et al. presented synthesis methods of stilbene-based dyes containing benzoxazole substituents and demonstrated improved solubility and fluorescence emission efficiency. Overall, the manuscript can be understood, but it is required to go through extensive editing of English language. There are a number of typos and grammatical errors in the manuscript. Also, there are some points need to be clarified.

(1) The authors can add some detailed explanations about why Stokes' shift occurred and varied with respect to the dyes.

(2) BBS with high concentration showed significant reduction in QY. However, PL peaks of BBS show narrower FWHM than that of other dyes. The authors need to explain why BBS has high FRET mechanism even though its FWHM is narrower. 

(3) In Figure 5, a picture of BBS can be added for the better comparison.

(4) QY of SBnAC has shown the higher QY than that of BBS. However, the solubility of SBnAC is not shown in Table 2. Solubility of SBnAC needs to be mentioned in Table 2.

Author Response

All authors would appreciate the valuable comments to our manuscript. Responding to the individual point, responses were shown in the revised manuscript in red color.

Response to Reviewer 1 comments :

Lee et al. presented synthesis methods of stilbene-based dyes containing benzoxazole substituents and demonstrated improved solubility and fluorescence emission efficiency. Overall, the manuscript can be understood, but it is required to go through extensive editing of English language. There are a number of typos and grammatical errors in the manuscript. Also, there are some points need to be clarified.

Point 1 : The authors can add some detailed explanations about why Stokes' shift occurred and varied with respect to the dyes.

Response 1 : The explanations included in the original manuscript, however as it was commented, more discussions were added in the discussion part of absorption and fluorescent properties. (p 12)

Point 2 : BBS with high concentration showed significant reduction in QY. However, PL peaks of BBS show narrower FWHM than that of other dyes. The authors need to explain why BBS has high FRET mechanism even though its FWHM is narrower. 

Response 2 : We agree. In the original manuscript, it was assumed that the FRET mechanism was triggered because the emission peak was red-shifted at high concentration. However, as it was commented, this was not enough to confirm FRET occurrence. Therefore, the manuscript was edited to delete the FRET word. (p 11)

Point 3 : In Figure 5, a picture of BBS can be added for the better comparison.

Response 3 : The picture was added in Figure 5.

Point 4 : QY of SBnAC has shown the higher QY than that of BBS. However, the solubility of SBnAC is not shown in Table 2. Solubility of SBnAC needs to be mentioned in Table 2.

Response 4 : The solubility of SBnAc was added in Table 2.

Reviewer 2 Report

The manuscript submitted by Choi and co-workers describe the synthesis and spectroscopic properties of six stilbene-based dyes. They aim at increasing the solubility of BBS in organic solvents and improve the quantum yield of these fluorescent molecules. Solubility in methylene chloride and quantum yields of the funtionalized BBS derivatives were significantly improved compared to the original BBS.

The manuscipt requires extense revision of the English language by a native speaker. I have also found several typograohical errors. Some examples are:

- tetrachloroethane (C2Cl4) = tetrachloroethane (C2H2Cl4)

- At the end of page 8: in in the range = in the range

- Luminance = luminiscence

In general the manuscript is fine and I can recommend its publication in applied science journal after extense language revision and considering the folowing issues:

1) The emission band of BSS shown sructured peaks characteristic of monomer emission. However, all the derivatives show a red-shift emission compared to the original BSS and very broad emission bands. I wonder if this has something to do with excimer emission. I suggest adding a comment on that and probably some experiments to confirm that indeed the emission is from the monomer.

2) I suggest changing the arrows in scheme 1 and 2 by "reaction arrows" and not using retrosynthesis arrows.

3) I suggest the authors to homogenize the units for concentration in molar units in the whole manuscript and avoid using ppm or g/L.

4) References are not well cited. The first name of the author appears instead of the last name. Please correct all of them.

Author Response

All authors would appreciate the valuable comments to our manuscript. Responding to the individual point, responses were shown in the revised manuscript in red color.

Response to Reviewer 2 comments:

The manuscript submitted by Choi and co-workers describe the synthesis and spectroscopic properties of six stilbene-based dyes. They aim at increasing the solubility of BBS in organic solvents and improve the quantum yield of these fluorescent molecules. Solubility in methylene chloride and quantum yields of the funtionalized BBS derivatives were significantly improved compared to the original BBS.

Point 1 : The manuscipt requires extense revision of the English language by a native speaker. I have also found several typograohical errors. Some examples are:

- tetrachloroethane (C2Cl4) = tetrachloroethane (C2H2Cl4)

- At the end of page 8: in in the range = in the range

- Luminance = luminescence

In general the manuscript is fine and I can recommend its publication in applied science journal after extense language revision and considering the following issues:

Response 1 : As it was commented, we corrected typographical errors and improved the English.

Point 2 : 1) The emission band of BSS shown structured peaks characteristic of monomer emission. However, all the derivatives show a red-shift emission compared to the original BSS and very broad emission bands. I wonder if this has something to do with excimer emission. I suggest adding a comment on that and probably some experiments to confirm that indeed the emission is from the monomer.

Response 2 : The graph in Figure was modified, as it was commented, but this graph only shows the shift of emission maximum depending on the dye concentrations of BBS dye, therefore it is not clear the emission type of the prepared six dyes was with respect to either a monomer emission or excimer emission.   

Point 3 : 2) I suggest changing the arrows in scheme 1 and 2 by "reaction arrows" and not using retrosynthesis arrows.

Response 3 : It was corrected in Schemes 1&2 by the reaction arrows.

Point 4 : 3) I suggest the authors to homogenize the units for concentration in molar units in the whole manuscript and avoid using ppm or g/L.

Response 4 : It was corrected using a molar unit to replace g/L or ppm unit.

Point 5 : 4) References are not well cited. The first name of the author appears instead of the last name. Please correct all of them.

Response 5 : As it was commented, the references were corrected.

Reviewer 3 Report

In this article, the authors discuss the synthesis and characterization of six new stilbene-based dyes with benzoxazole substituents to improve their solubility and fluorescence efficiency at blue emission. The authors investigated the absorption and fluorescence properties of the synthesized dyes, which were achieved by introducing -NO2 and -NH2 groups onto benzoxazolyl rings during the derivatization of benzoxazolyl stilbene dye. The emission maxima of the six dyes were observed in the range of 435~472 nm, which were red-shifted to the starting material. Additionally, the authors examined the solubility of the dyes in dichloromethane and found that N-alkyl groups had greater solubility compared to the analogue containing an unsubstituted group. The study also investigated the optical effects of tortional strain according to the conformation of the side group, which was analyzed by calculating the geometry, dihedral angle, and other parameters of the synthesized dyes using density functional theory and time-dependent function with Gaussian 09 program. Overall, the topic of this study is interesting and appropriate for the scope of the journal, and the experimental work was competently performed. With minor revisions, this work could be considered for publication.

1.     It is necessary for the author to include the 13C NMR data of the new compounds, compare the expected and observed mass of the compound, and provide the NMR spectra in the SI.

2.     In Page 10, “This effect has been reported in the literature………properties of dyes.” The references below clearly related to impact of alkyl substituents on the fluorescence properties of the dyes and had better be added into the revised manuscript. Achalkumar, et.al. J. Mater. Chem. C, 2016,4, 6546-6561, Balaram et.al. J. Mater. Chem. C, 2016,4, 6117-6130, Chapman et.al. Anal Chem Insights 2011; 6: 29–36.

3.     The author should provide the solid-state fluorescence spectra for all the dyes and compare them with the spectra obtained from dilute solutions. Additionally, they should attempt to explain the type of aggregation present, whether it is H-type or J-type.

Author Response

All authors would appreciate the valuable comments to our manuscript. Responding to the individual point, responses were shown in the revised manuscript in red color.

Response to Reviewer 3 comments:

In this article, the authors discuss the synthesis and characterization of six new stilbene-based dyes with benzoxazole substituents to improve their solubility and fluorescence efficiency at blue emission. The authors investigated the absorption and fluorescence properties of the synthesized dyes, which were achieved by introducing -NO2 and -NH2 groups onto benzoxazolyl rings during the derivatization of benzoxazolyl stilbene dye. The emission maxima of the six dyes were observed in the range of 435~472 nm, which were red-shifted to the starting material. Additionally, the authors examined the solubility of the dyes in dichloromethane and found that N-alkyl groups had greater solubility compared to the analogue containing an unsubstituted group. The study also investigated the optical effects of tortional strain according to the conformation of the side group, which was analyzed by calculating the geometry, dihedral angle, and other parameters of the synthesized dyes using density functional theory and time-dependent function with Gaussian 09 program. Overall, the topic of this study is interesting and appropriate for the scope of the journal, and the experimental work was competently performed. With minor revisions, this work could be considered for publication.

Point 1 :   It is necessary for the author to include the 13C NMR data of the new compounds, compare the expected and observed mass of the compound, and provide the NMR spectra in the SI.

Response 1 : It was modified to compare the calculated and observed mass and 1H NMR as well (5~7 p.). However for 13C NMR, it takes too long times, dye synthesis, purification by column chromatography, and NMR analysis from outside our research lab those will exceed the deadline of revision (within 10 days after reviewers’ comments delivered). Normally it is acceptable by 1H NMR and MASS for the publication.

Comment 2 :  In Page 10, “This effect has been reported in the literature………properties of dyes.” The references below clearly related to impact of alkyl substituents on the fluorescence properties of the dyes and had better be added into the revised manuscript. Achalkumar, et.al. J. Mater. Chem. C, 2016,4, 6546-6561, Balaram et.al. J. Mater. Chem. C, 2016,4, 6117-6130, Chapman et.al. Anal Chem Insights 2011; 6: 29–36.

Response 2 : We agree, all the commented references were added.

Comment 3 :  The author should provide the solid-state fluorescence spectra for all the dyes and compare them with the spectra obtained from dilute solutions. Additionally, they should attempt to explain the type of aggregation present, whether it is H-type or J-type.

Response 3 : We also thought that we should obtain solid-state quantum yield data, however the quantum yield of the solid state could not be measured with our lab’s equipment. Instead, we added additional discussion about aggregation type (14 p.).

Round 2

Reviewer 1 Report

The authors have answered all the questions raised and thoroughly revised the manuscript. It is believed that the manuscript is ready to be published.

Author Response

There are several errors with regard to the English, as for example.

Comment 1) Abstract

-Besides, the solubility of the dyes in dichloromethane was examined for applicating to the nonpolar polymer films such as PE, PP, PVC and so on. It should be “for the application” or “for applying”.

-Alkylated should not be capitalised.

-parameters of THE synthesized dyes

Keywords

blue fluorescent should blue fluorescence.

In caption of fig. 4 fluorescent should be emission

Therefore, the English must be revised by a mother tongue.

Answer :  It’s done by correcting the abstract.

Comment 2) Also, there are details to be fixed, for example C28H16N4O6 in 2.3.1. Synthesis of dye intermediate 1: nitration. You should add the subscripts in all the formular.

Answer :  All formula changed to be subscript format.

 Comment 3) It is not sufficiently clear the red shift observed in Figure 4. Please add more details.

Answer : It’s done. More discussions were added on 14 page along with the relevant references of 25~27.
